# Natural History from Appendiceal Mucocele to Jelly Belly

**DOI:** 10.3390/diagnostics14222532

**Published:** 2024-11-12

**Authors:** David Hoskovec, Zdeněk Krška, Adam Pudlač, Matyáš Lochman, Sabina Strohalmová, Andrej Bocán, Pavel Koželský, Petr Dytrych

**Affiliations:** 11st Department of Surgery, General University Hospital Prague, 128 00 Prague, Czech Republic; zdenek.krska@vfn.cz (Z.K.); matyas.lochman@vfn.cz (M.L.); sabina.strohalmova@vfn.cz (S.S.); pavel.kozelsky@vfn.cz (P.K.); petr.dytrych@vfn.cz (P.D.); 21st Faculty of Medicine, Charles University, 120 00 Prague, Czech Republic; adam.pudlac@vfn.cz (A.P.); andrej.bocan@vfn.cz (A.B.); 3Departments of Radiodiagnostics, General University Hospital Prague, 128 00 Prague, Czech Republic

**Keywords:** mucocele, pseudomyxoma peritonei, jelly belly, HIPEC, cytoreductive surgery

## Abstract

Mucocele of the appendix and pseudomyxoma peritonei are rare diseases. The clinical findings are nonspecific in the early stages of the disease. The sequelae of appendiceal mucocele, its perforation, and extensive peritoneal involvement via pseudomyxoma peritonei (jelly belly) are repeatedly described in the literature. We present the typical findings in the natural history of the disease.

Pseudomyxoma peritonei is an extremely rare condition. Its incidence is about 1–4 cases/1 million/year [1,2,3,4]. There is a slightly higher incidence in females than in males (11:9) [5]. Most of these develop after a rupture of the appendicular mucocele. Appendicular mucoceles are uncommon with an incidence of about 0.2–0.7% of the population [4,6]. Progression to pseudomyxoma peritonei develops in 20% of patients [4]. Appendicular tumors (independent of histological findings) are found in 1% of all appendectomy specimens.

Appendiceal mucocele

Mucocele of the appendix refers to a dilated mucin-containing appendix. This term is mainly used by clinicians and radiologists (Figure 1, Figure 2, Figure 3 and Figure 4). It is necessary to distinguish “true mucocele” from a mucinous adenoid neoplasm [7]. The term mucocele was first used by Karl Freiherr von Rokitansky in 1842. Two years later, Werth used the term pseudomyxoma peritonei [4]. The clinical signs are minimal or often completely absent. The rupture of the mucocele can also be asymptomatic. Mucocele may be found on ultrasound or via a regular gynecological examination. A CT scan confirms the diagnosis. Treatment consists of the surgical removal of the dilated appendix, often together with the surrounding part of the cecum (Figure 5, Figure 6 and Figure 7). This approach is curative if there is no rupture or presence of tumor cells on the peritoneal surface. The diagnosis is confirmed through pathological examination of the specimen (Figure 8).

Pseudomyxoma peritonei

Pseudomyxoma peritonei is a clinical diagnosis characterized by the accumulation of mucinous masses inside the abdominal cavity. It occurs as a result of a rupture of the appendiceal mucocele. The progression from mucocele to pseudomyxoma usually takes several years.

Mucinous implantation and accumulation lead to abdominal distension, discomfort, pain, and bowel obstruction. A palpable mass in the abdomen is a late symptom [4]. The localization of mucin and a tumor is characterized by the phenomenon of redistribution [8]. This means that the tumor cells move with the intra-abdominal fluid and also according to gravity. The highest concentration is at the gravity gradient sites and then at the sites of fluid absorption from the abdominal cavity.

The treatment for pseudomyxoma peritonei is based on the Sugarbaker technique of cytoreductive surgery and hyperthermic intraperitoneal chemotherapy (CRS + HIPEC), sometimes followed by early postoperative intraperitoneal chemotherapy (EPIC) for five days. The results of this treatment are very favorable. The 5-year survival rate predicted via the Kaplan–Meier method is over 80% [8].

In this article, we show the natural history of the disease and the typical findings at the different stages of this rare diagnosis.

**Figure 1 diagnostics-14-02532-f001:**
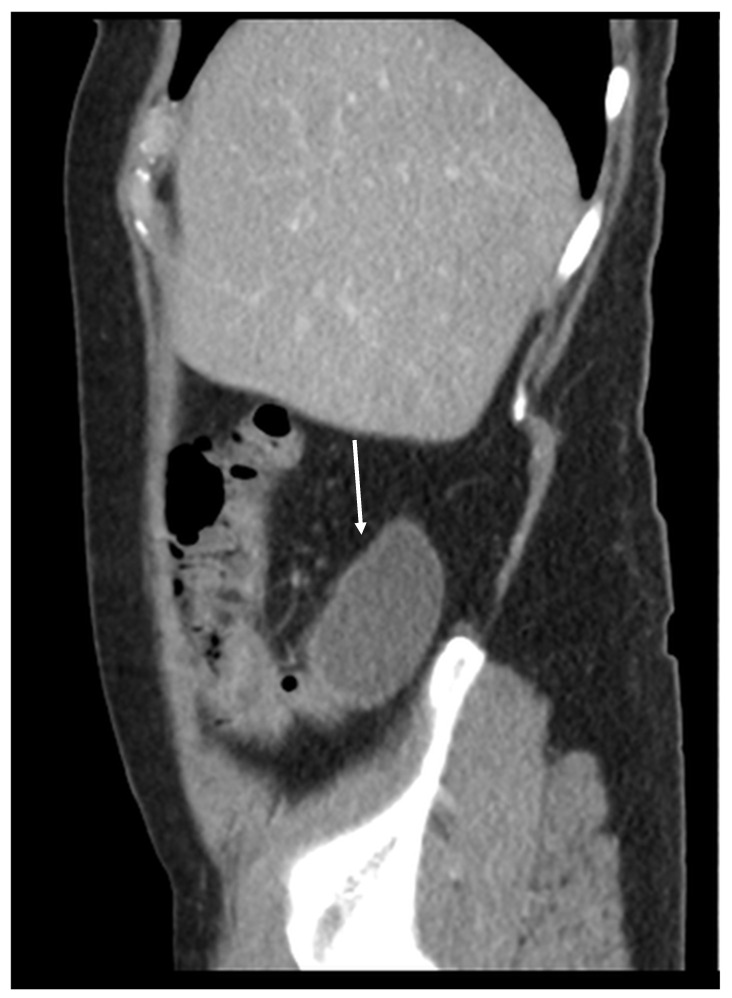
Mucocele of the appendix (arrow) (CT, venous phase, sagittal plane). There is cystic dilatation of the appendix behind the ascending colon. There is no evidence of perforation or the presence of mucin outside the mucocele.

**Figure 2 diagnostics-14-02532-f002:**
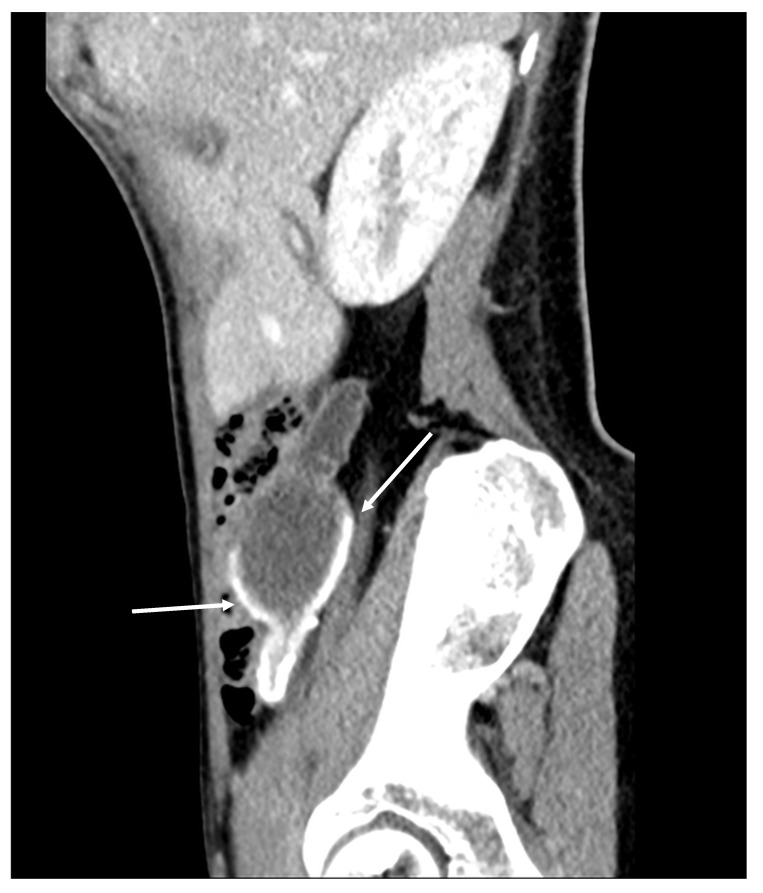
Mucocele of the appendix with calcification in the mucocele’s wall (arrows) (CT, venous phase, sagittal plane). Cystic dilatation of the appendix and calcification in the appendiceal wall. This is a typical sign for a mucocele, and it is present in 50% of patients [9].

**Figure 3 diagnostics-14-02532-f003:**
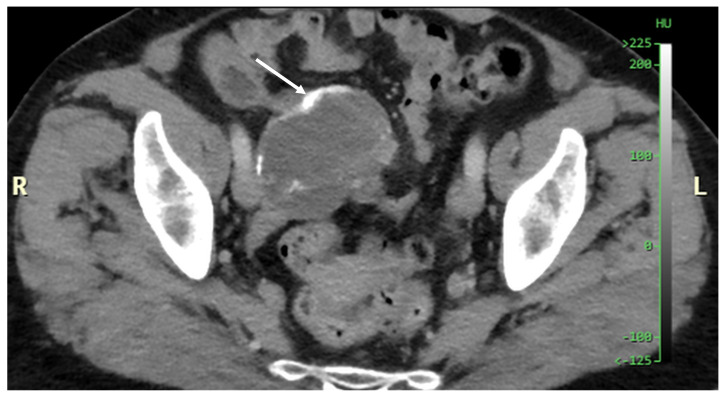
Another example of an appendiceal mucocele with calcification in the wall (arrow) (CT, venous phase, axial plane).

**Figure 4 diagnostics-14-02532-f004:**
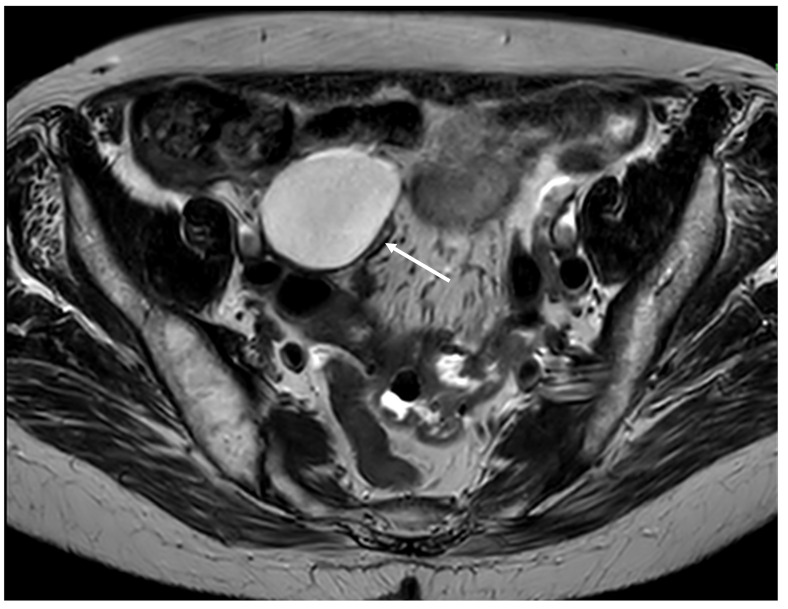
Cystic dilatation of the appendix (mucocele) (arrow) (MRI, axial plane).

**Figure 5 diagnostics-14-02532-f005:**
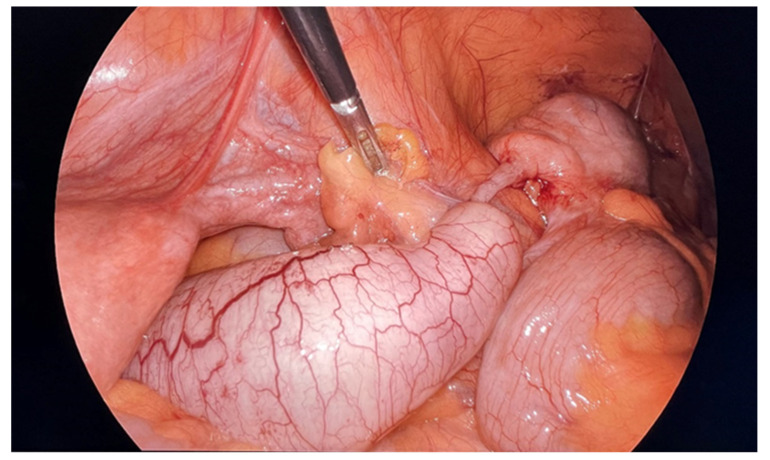
Mucocele of the appendix without perforation during laparoscopic surgery. There is a partial dilatation of the appendix and a healthy appendicular base.

**Figure 6 diagnostics-14-02532-f006:**
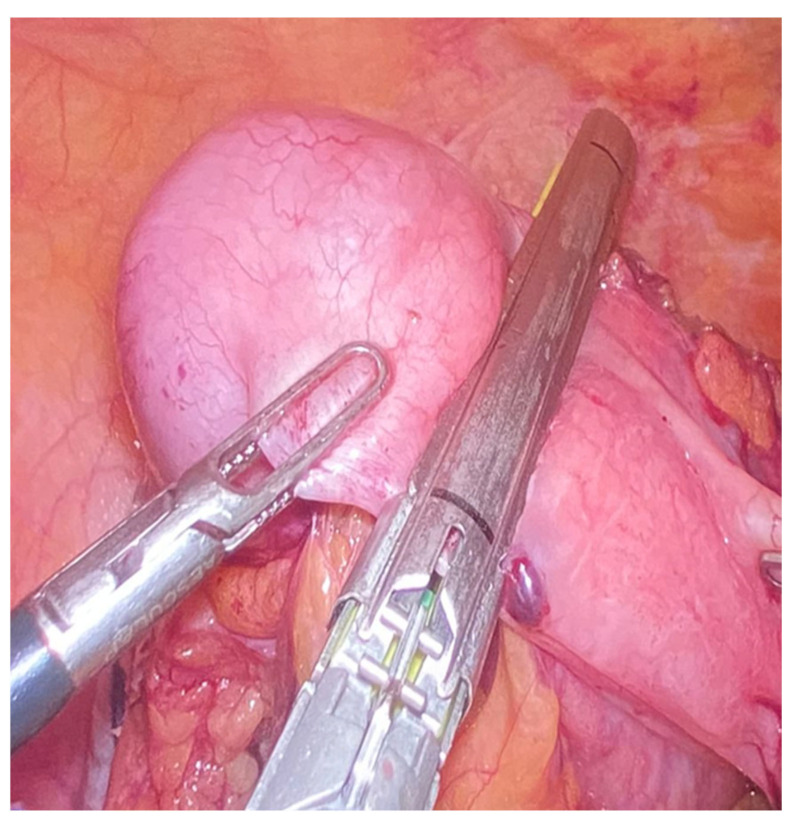
Radical appendectomy including appendiceal mesentery and cecal bases with linear stapler.

**Figure 7 diagnostics-14-02532-f007:**
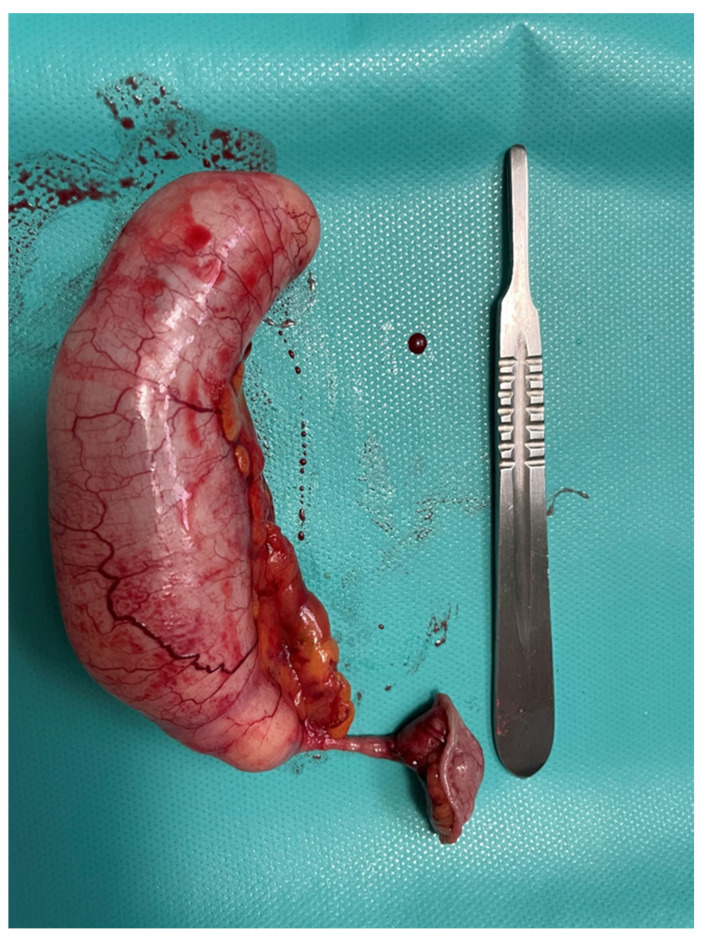
The specimen.

**Figure 8 diagnostics-14-02532-f008:**
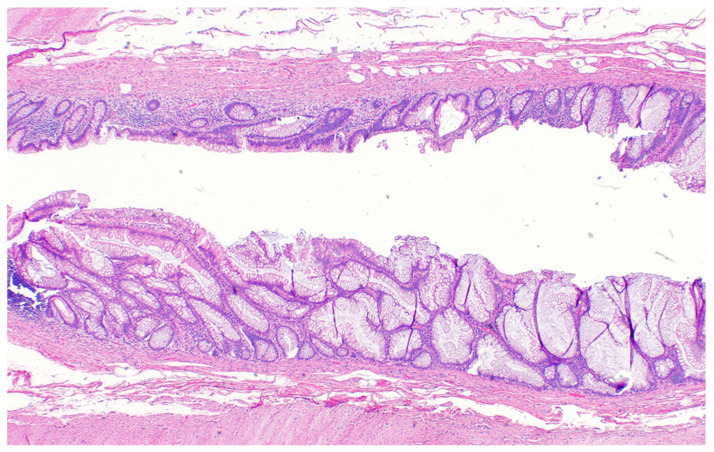
Pathological specimen (40×). The wall of the appendix is intact, there is no sign of perforation and no signs of malignancy, and mucin is only in the appendicle lumen.

The mucocele ruptures due to increased intra-appendicular pressure, releasing mucus and tumor cells into the peritoneal cavity (Figure 9 and Figure 10). The rupture may be asymptomatic or associated with only mild and nonspecific clinical signs. Typical peritoneal signs and bacterial peritonitis are usually absent because the communication to the lumen of the colon is closed by mucin. Treatment in these cases is called a radical appendectomy, with the resection of the cecal bases, but without a right colectomy. The addition of HIPEC is fully indicated. It is necessary to examine the whole abdomen to exclude the presence of tumors in typical locations, such as the Douglas pouch and the undersurfaces of the diaphragm and the right and left paracolic gutter.

The current pathophysiological mechanism is based on the Sugarbaker theory. The adenoma forms in the lumen of the appendix, and as it grows, the lumen becomes occluded. As a result of this occlusion, a distended part of the appendix dilates, and the wall of the appendix ruptures. The mucous membrane and epithelial cells from the adenoma are then released and shed into the abdominal cavity [8].

Patients with a misdiagnosed rupture of the mucocele may develop pseudomyxoma peritonei. It is characterized by the presence of an abundant gelatinous substance in the abdomen. Diffuse, progressive, and abundant mucin-containing tumor cells are typical of this disease [2] (Figure 11, Figure 12, Figure 13, Figure 14, Figure 15 and Figure 16). The interval between rupture and advanced disease is several years. In our clinical series, the interval was about 5 years. In the literature, we found a wide range of this interval from 12 months to 10 years [9,10]. The natural history of PMP revolves around the “redistribution phenomenon”, whereby mucinous tumor cells accumulate in the Douglas pouch, in the diaphragm (more on the right), and the small and large omentum (Figure 17). The small intestine is less involved [4]. Pseudomyxoma peritonei is a slowly progressing disease, which fills the peritoneal cavity over time. There are several classifications of pseudomyxoma. The most commonly used is the PSOGI classification [2]. Mucinous accumulation progresses to malnutrition, bowel obstruction, and respiratory compromise. Rarely, the tumor may spread to the pleural cavity. This has been described in 5.4% of cases. It may occur spontaneously or as a result of diaphragmatic injury during cytoreduction [8,11].

Figure 11, Figure 12, Figure 13, Figure 14 and Figure 15. Advanced stage of the disease. The abdomen is full of tumors and mucin. Typical signs are heterogeneous or hypodense masses in the form of lobules often with septa, which could be enhanced by contrast and could be associated with calcifications.

Treatment of the advanced disease consists of maximal cytoreduction (Figure 18 and Figure 19), which means the removal of all visible tumors and HIPEC. The abdominal cavity is lavaged with a warmed cytostatic solution at 42 degrees Celsius for 90 min. The most commonly used cytostatic is mitomycin. The clinical classification of pseudomyxoma peritonei is based on the Sugarbaker peritoneal cancer index, which can stratify the extent of the disease. The radicality of surgery is assessed according to the cytoreduction score [3,8,12].

Patients who have undergone cytoreductive surgery and HIPEC due to pseudomyxoma peritonei should be followed up with imaging studies at regular intervals. Recurrence may occur after several years (Figure 20 and Figure 21).

Conclusions

Pseudomyxoma peritonei is a rare disease that occurs as a result of perforation of the appendiceal mucocele, which is also rare. It is necessary to know the typical clinical, imaging, and operative findings. The treatment follows the recommendations of PSOGI. Overtreatment should be avoided in the case of uncomplicated findings without perforation; on the other hand, early referral of patients with a perforated mucocele to a center with the possibility of HIPEC treatment will prevent extensive disease.

In this article, we showed the typical findings in the early and advanced stages of the disease. A limitation is the wide clinical variability, especially during disease progression, where not all possibilities can be covered herein.

## Figures and Tables

**Figure 9 diagnostics-14-02532-f009:**
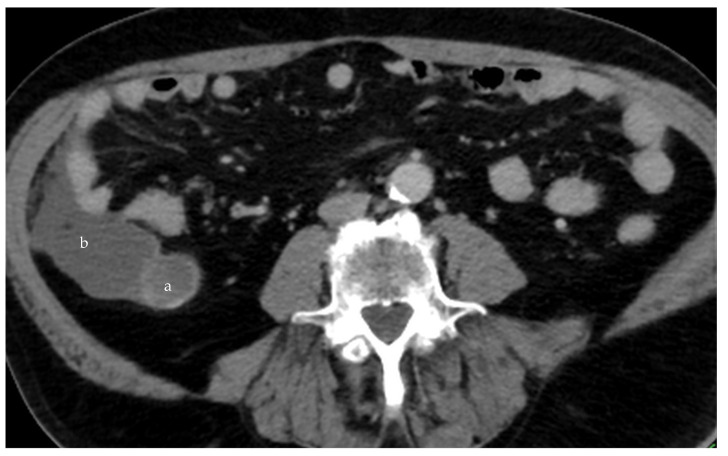
Ruptured mucocele of the appendix (**a**), with free mucin around the appendix (**b**) (CT, venous phase, axial plane).

**Figure 10 diagnostics-14-02532-f010:**
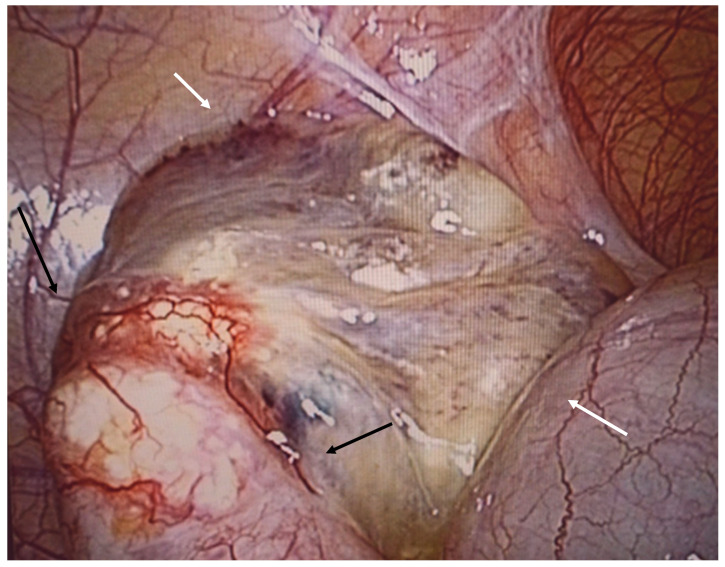
Ruptured mucocele of the appendix in perioperative view. Part of the appendix (black arrow) is embedded in mucus (white arrow), which covers the perforation.

**Figure 11 diagnostics-14-02532-f011:**
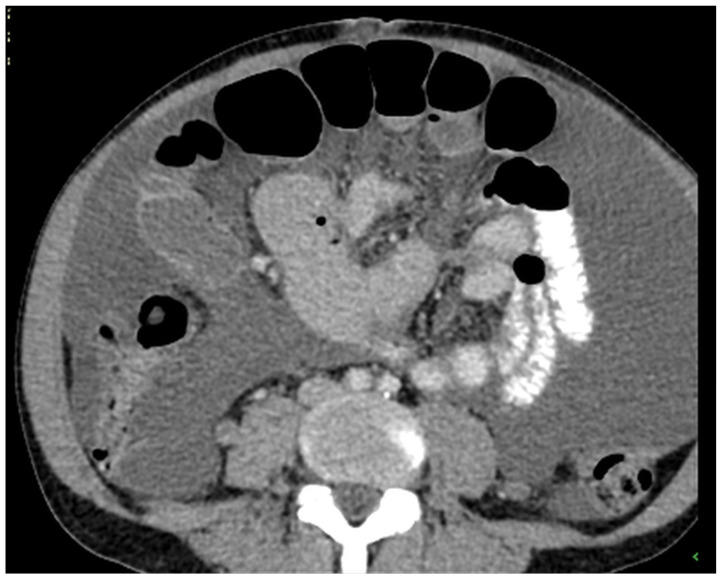
CT, venous phase, axial plane.

**Figure 12 diagnostics-14-02532-f012:**
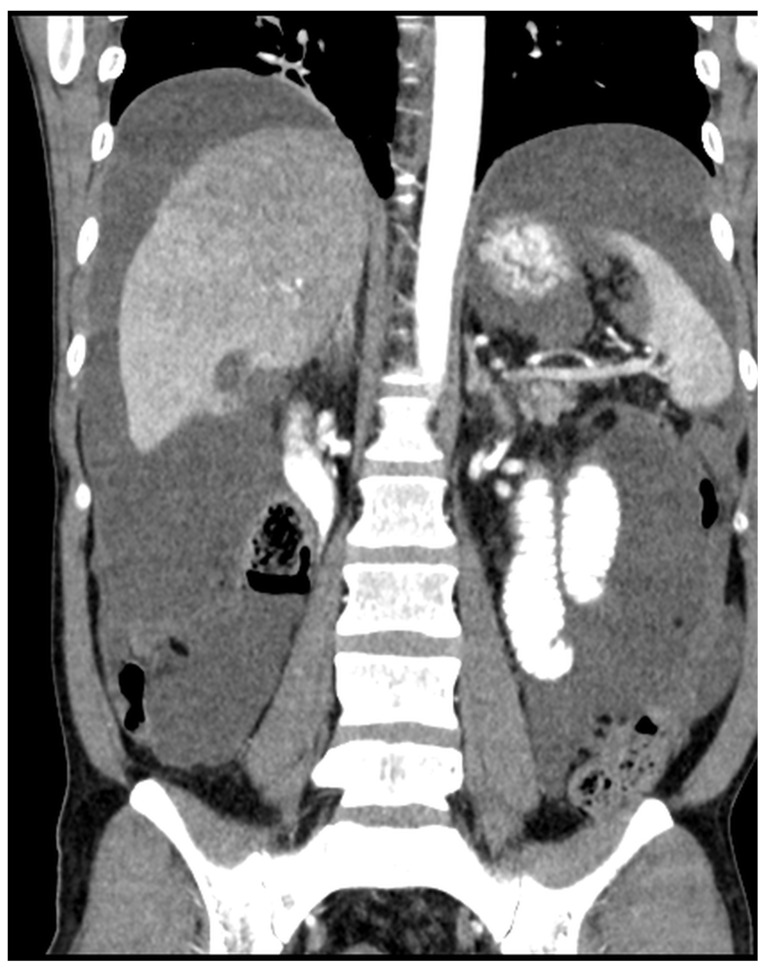
CT, venous phase, coronar plane.

**Figure 13 diagnostics-14-02532-f013:**
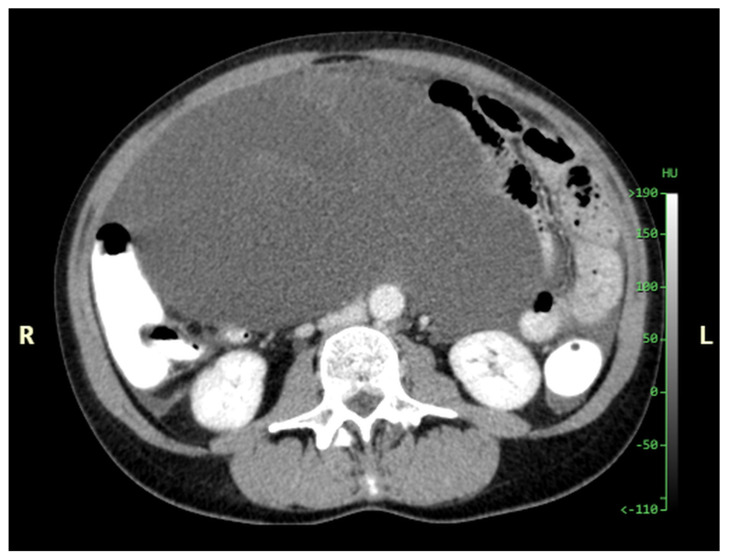
CT, venous phase, axial plane.

**Figure 14 diagnostics-14-02532-f014:**
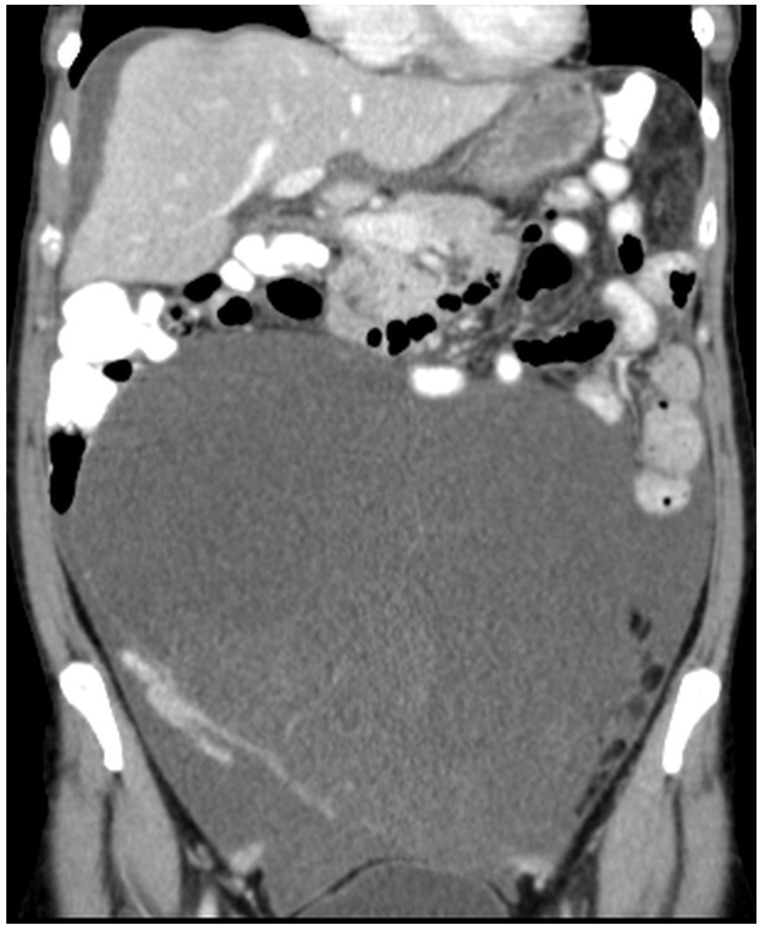
CT, venous phase, coronar plane.

**Figure 15 diagnostics-14-02532-f015:**
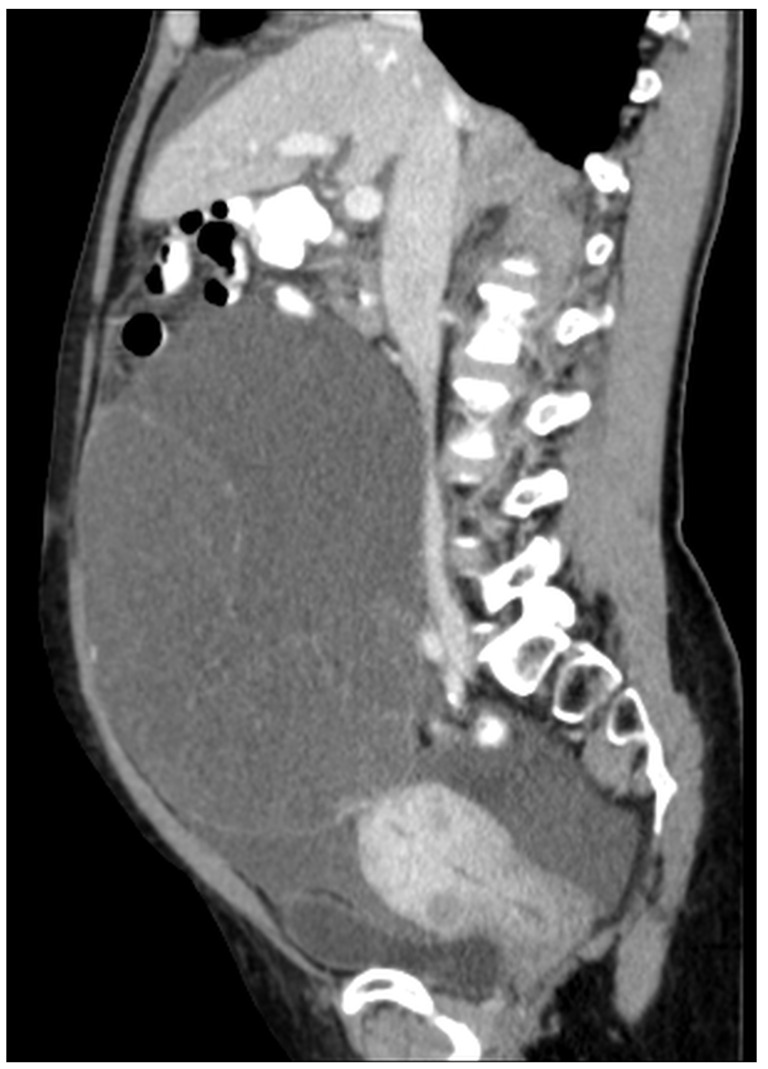
CT, venous phase, sagittal plane.

**Figure 16 diagnostics-14-02532-f016:**
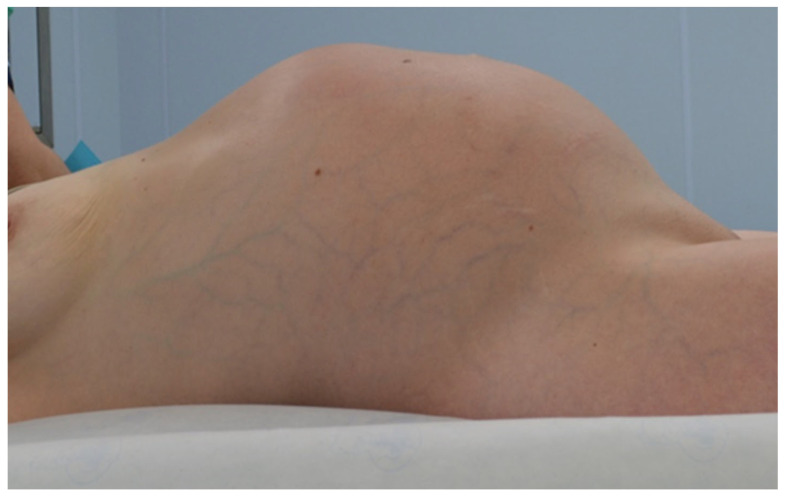
Distended abdomen due to a large accumulation of mucin and tumorous masses.

**Figure 17 diagnostics-14-02532-f017:**
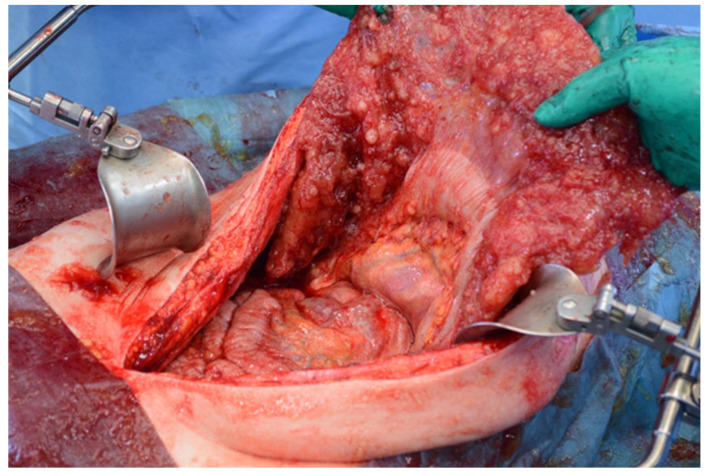
Intraoperative view of the abdominal cavity. The omentum contains many tumors.

**Figure 18 diagnostics-14-02532-f018:**
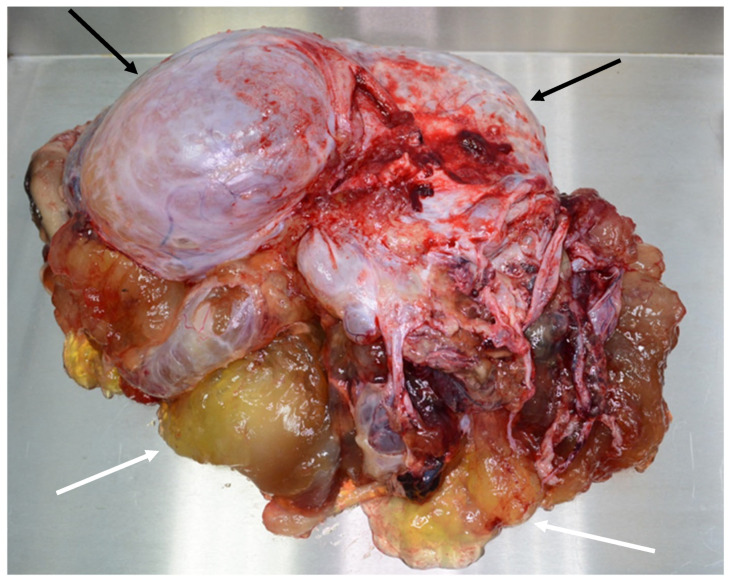
Specimen with tumor (black arrows) and mucin (white arrows).

**Figure 19 diagnostics-14-02532-f019:**
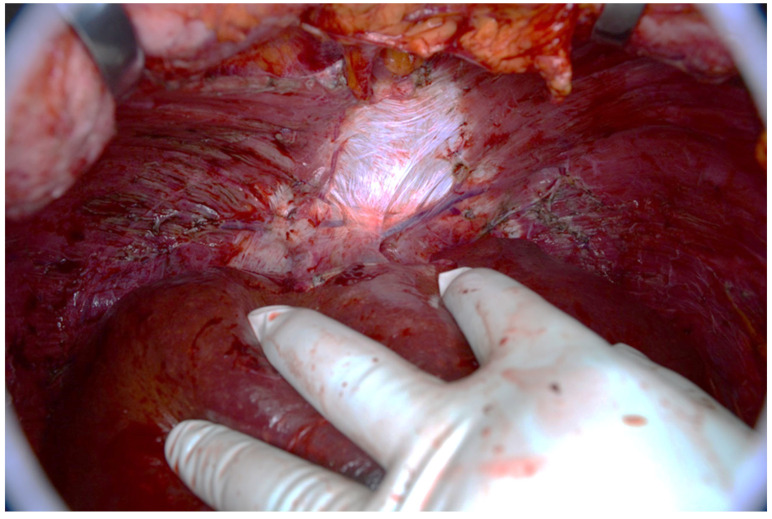
The diaphragm after removing the peritoneum with tumors.

**Figure 20 diagnostics-14-02532-f020:**
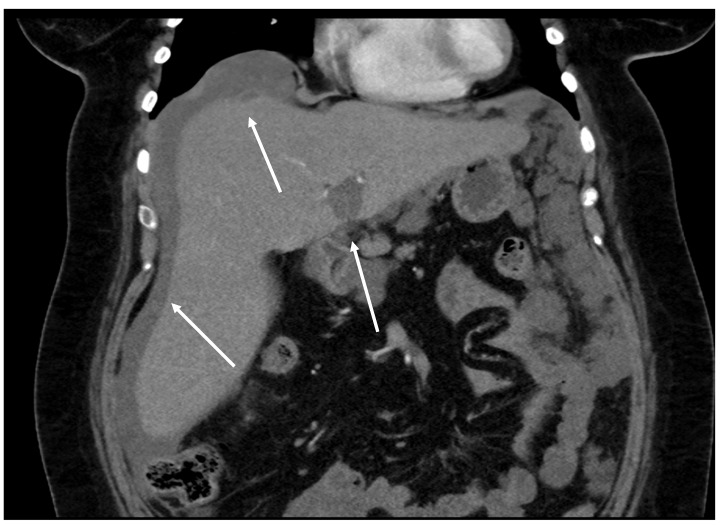
Recurrence four years after CRS + HIPEC. Mucinous tumor around the liver and in the porta hepatis (arrows). This patient was successfully treated with repeated CRS + HIPEC. (CT, portal phase, coronar plane).

**Figure 21 diagnostics-14-02532-f021:**
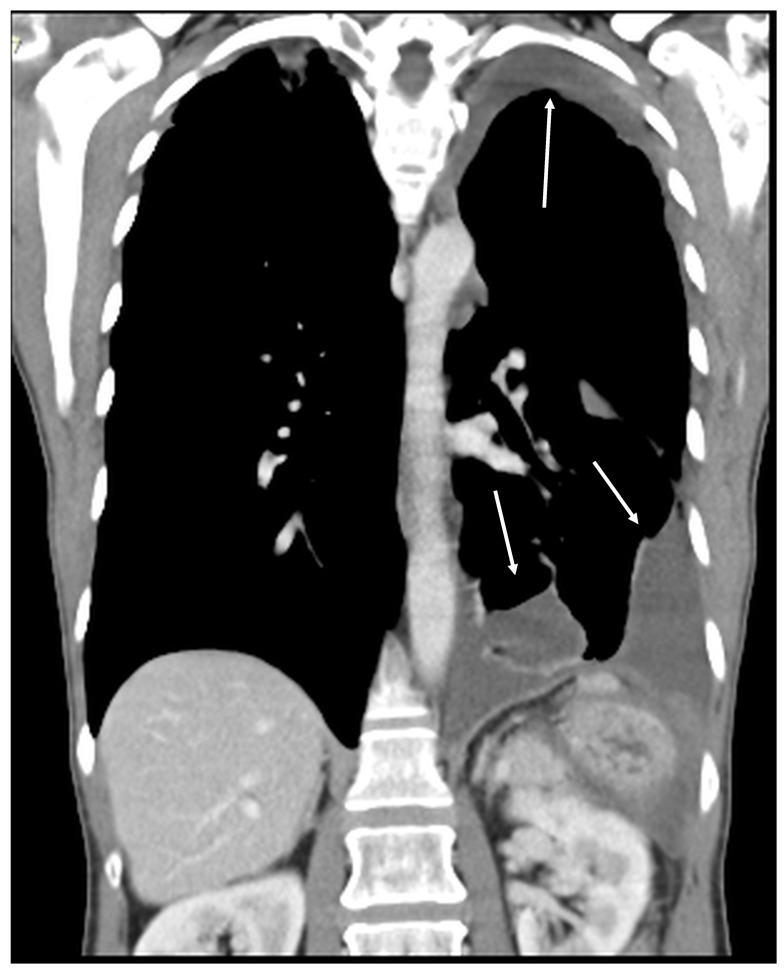
Recurrence of the pseudomyxoma peritonei in the abdominal and left thoracic cavities (arrows). The patient underwent two operations: first, CRS + HIPEC to address the intra-abdominal disease; then, CRS + HITHOC in the left thoracic cavity. The cause of the spread of the disease to the chest would probably be a thermal lesion of the diaphragm, which manifested sometime after the primary surgery. (CT, venous phase, coronar plane).

## Data Availability

No new data were created or analyzed in this study.

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
