# Peer review of "Natural History from Appendiceal Mucocele to Jelly Belly"

_diagnostics, 2024, doi:10.3390/diagnostics14222532_

Round 1
Reviewer 1 Report
Comments and Suggestions for Authors
1. Type of Article
1.1. Interesting Images – Need to reconsider.
2. Title
2.1. It should be appendiceal instead of “appendical”.
3. Abstract
3.1. Revise and replace “clear and well known” with more scientific terms.
4. Introduction (Perhaps)
4.1. There is a slightly higher incidence in women (9:11) is vague. Please revise.
4.2. Try not to repeat and rephrase similar statements regarding the incidence of pseudomyxyxoma peritonei.
4.3. Review and revise “The localization of mucin and tumour is characterized by the phenomenon of redistribution”. It is vague.
4.4. Check and revise: “However, EPIC has been abandoned in many centers in recent years” immediately followed by “The results of this treatment are very favorable” and a predicted 5-year survival rate over 80%.
4.5. The last paragraph should be moved up and remove repetition.
5. Figures (Perhaps case presentation, results or discussion section)
5.1. Each figure must have been cited previously in the text.
5.2. Describe the type of CT scan and view. Similarly, the type of MRI.
5.3. Add arrows to point at least pathologies to guide readers in all the figures.
5.4. Figure 1: Leave out “clearly visible”.
5.5. Correct “subfrenae” Lune 82-83.
5.6. Figure 10 is not clear. Classification of PMP in Line 104-105 should have been covered in the introduction.
5.7. Caption/Legend for each of Figure 11-15 should be presented separately.
5.8. Figure 17: It should be intra-operative instead of peroperative.
5.9. Figure 18: The caption/legend is vague.
5.10. HIPEC has already been defined and used. Need to use it consistently (Linex143-144).
6. Conclusion: Noted.
7. General comment
7.1. The manuscript needs to be structured.
7.2. Captions or legends of the figures must be below each.
7.3. Sources of the pictures need to be acknowledged.

Although the quality is reasonable, English language requires revision and improvement.
Author Response
Dear reviewer
Thank you for your time and valuable comments on our article. We have done our best to respond to your comments. The article has been edited and partially reordered according to your recommendations and those of your colleagues. The article has also undergone English language editing by MDPI.
Your sincerely
- H.
- Type of Article
1.1. Interesting Images – Need to reconsider.
Our study focuses on the evolution of this rare disease supported by imaging and clinical documentation, so we believe that Interesting Images is the correct choice for our article
- Title
2.1. It should be appendiceal instead of “appendical”.
Corrected
- Abstract
3.1. Revise and replace “clear and well known” with more scientific terms.
Corrected
- Introduction (Perhaps)
4.1. There is a slightly higher incidence in women (9:11) is vague. Please revise.
We used a new, more precise expression
4.2. Try not to repeat and rephrase similar statements regarding the incidence of pseudomyxyxoma peritonei.
Corrected
4.3. Review and revise “The localization of mucin and tumour is characterized by the phenomenon of redistribution”. It is vague.
We have tried to clarify the term and describe it more precisely
4.4. Check and revise: “However, EPIC has been abandoned in many centers in recent years” immediately followed by “The results of this treatment are very favorable” and a predicted 5-year survival rate over 80%.
We abandoned the sentence about EPIC, it is not so important and in this sentence order can be confusing
4.5. The last paragraph should be moved up and remove repetition.
We completely rearranged the paragraphs.
- Figures (Perhaps case presentation, results or discussion section)
5.1. Each figure must have been cited previously in the text.
We followed the guidelines for Authors of Interesting Images, where the article is a series of images with a broader description. Therefore, the images are not mentioned in the text, but we consider them to be the main part of the paper.
5.2. Describe the type of CT scan and view. Similarly, the type of MRI.
We added it
5.3. Add arrows to point at least pathologies to guide readers in all the figures.
We added arrows to figures to improve the understanding of the pathology
5.4. Figure 1: Leave out “clearly visible”.
The expression was removed
5.5. Correct “subfrenae” Lune 82-83.
Corrected
5.6. Figure 10 is not clear.
We tried to improve it by arrows
Classification of PMP in Line 104-105 should have been covered in the introduction.
The classification was in first version of the article, but it was removed after preliminary check by Section Managing Editor Diagnostics
5.7. Caption/Legend for each of Figure 11-15 should be presented separately.
We added information about CT scans
5.8. Figure 17: It should be intra-operative instead of peroperative.
Corrected
5.9. Figure 18: The caption/legend is vague.
We added some arrows to better understanding
5.10. HIPEC has already been defined and used. Need to use it consistently (Linex143-144).
Corrected
- Conclusion: Noted.
- General comment
7.1. The manuscript needs to be structured.
7.2. Captions or legends of the figures must be below each.
7.3. Sources of the pictures need to be acknowledged.
The article is not structured like classical scientific paper, because it is not required when submitting an article to Interesting Images section.
The images are the property of the hospital or the authors of the article, except for the microscopic finding, where acknowledgement is given
Reviewer 2 Report
Comments and Suggestions for Authors
i have read the manuscript no 3274691
it is about mucocele and pseudomyxoma peritonei.
Although it is an interesting manuscript with many operative and diagnostic imaging pictures and although it is interesting to study the treatment of these two different conditions i have some objections to the structure of the text because it confuses the reader.
i propose it would be preferable apart from the first paragraph where the two entities are mentioned together in order to explain the aim of the paper, to present each one separetely including its treatment, because in the present structure the authors jump from the one condition to another confusing the reader.
i propose to authors to make if possible these changes in order to accept their manuscript
thank you in advance
Author Response
Dear reviewer
Thank you for your time and valuable comments on our article. We have done our best to respond to your comments. The article has been edited and partially reordered according to your recommendations and those of your colleagues. Especially the first paragraph was completely rearranged. The article has also undergone English language editing by MDPI.
Your sincerely
D.H.
Reviewer 3 Report
Comments and Suggestions for Authors
Dear authors, your paper touches an interesting topic.
I have the suggestions for you:
ABSTRACT
The abstract fully encapsulates the study objectives, main findings, and relevance. Highlight what your article adds to the existing literature on PMP to underscore its value.
INTRODUCTION
I’d like to expand briefly on the clinical importance of PMP and appendiceal mucocele detection and treatment options to capture reader interest immediately.
DISCUSSION
- Emphasize the study’s contribution to the understanding of PMP treatment outcomes. You should discuss the finding of your paper compared with other papers by literarure (add references).
CONCLUSION
Add a paragraph with the limit of the study.
Comments on the Quality of English Language
GRAMMATICAL REVIEW
You should correct minor typographical errors (e.g., “apendix” should be “appendix”) throughout, especially in figure captions. Consistency in medical terminology, like “mucinous adenoma” vs. “mucinous adenoid neoplasm,” is also recommended. A review by a native speaker is necessary.
Author Response
Dear reviewer
Thank you for your time and valuable comments on our article. We have done our best to respond to your comments. The article has been edited and partially reordered according to your recommendations and those of your colleagues. Especially the first paragraph was rearranged. There is not discussion because we followed the Instruction for Authors of Interesting Images (No regular manuscript text introduction/methods/results/discussion) should be included). We added sentence about limitation of the study. The article has also undergone English language editing by MDPI.
Your sincerely
D.H
Round 2
Reviewer 1 Report
Comments and Suggestions for Authors
Thank you very much to the authors for the revised manuscript. I am satisfied with all the explanations and changes made. The manuscript reads well.
Reviewer 3 Report
Comments and Suggestions for Authors
Now the paper is suitable to be publish in this journal.